# High-Quality Object Detection Method for UAV Images Based on Improved DINO and Masked Image Modeling

Wanjie Lu [1],*, Chaoyang Niu [1], Chaozhen Lan [1], Wei Liu [1], Shiju Wang [1], Junming Yu [2] and Tao Hu [1]

[1] Institute of Data and Target Engineering, PLA Strategic Support Force Information Engineering University, Zhengzhou 450001, China; niucy2017@outlook.com (C.N.); lan_cz@163.com (C.L.); greatliuliu@163.com (W.L.); 1733150660@139.com (S.W.); hutaoengineering@163.com (T.H.)

[2] 27th Research Institute, China Electronic Technology Group Corporation, Zhengzhou 450047, China; junmingy@163.com

* Correspondence: lwj285149763@163.com

**Abstract:** The extensive application of unmanned aerial vehicle (UAV) technology has increased academic interest in object detection algorithms for UAV images. Nevertheless, these algorithms present issues such as low accuracy, inadequate stability, and insufficient pre-training model utilization. Therefore, a high-quality object detection method based on a performance-improved object detection baseline and pretraining algorithm is proposed. To fully extract global and local feature information, a hybrid backbone based on the combination of convolutional neural network (CNN) and vision transformer (ViT) is constructed using an excellent object detection method as the baseline network for feature extraction. This backbone is then combined with a more stable and generalizable optimizer to obtain high-quality object detection results. Because the domain gap between natural and UAV aerial photography scenes hinders the application of mainstream pre-training models to downstream UAV image object detection tasks, this study applies the masked image modeling (MIM) method to aerospace remote sensing datasets with a lower volume than mainstream natural scene datasets to produce a pre-training model for the proposed method and further improve UAV image object detection accuracy. Experimental results for two UAV imagery datasets show that the proposed method achieves better object detection performance compared to state-of-the-art (SOTA) methods with fewer pre-training datasets and parameters.

**Keywords:** UAV image; object detection; masked image modeling; global–local hybrid

## 1. Introduction

With the widespread application of unmanned aerial vehicle (UAV) imaging technology, object detection methods capable of obtaining information from UAV images have provided services for urban planning, environmental monitoring, and disaster rescue and significantly reduced the need for human, material, and financial resources. Many UAV image object detection methods have been developed, and convolutional neural network (CNN)-based methods have become mainstream [1–3]. Methods based on CNNs mainly include two-stage methods that present high accuracy (such as the region-based CNN (R-CNN) series [4–9]) and one-stage methods that present high speed (you only look once (YOLO) series [10], single shot detector (SSD) [11], RetinaNet [12], and fully convolutional one-stage object detector (FCOS) [13]). With the emergence of transformers, especially visual transformers (ViTs) [14], an increasing number of object detection methods have been proposed based on the attention mechanism [15], such as detection transformer (DETR) [16], deformable DETR [17], denoising DETR (DN-DETR) [18], dynamic anchor box DETR (DAB-DETR) [19], shifted window transformer (Swin Transformer) [20], and pyramid vision transformer (PVT) [21,22].

CNN- and ViT-based object detection methods have shown strong performances in different scenarios; nevertheless, the following challenges remain [23–26]: (1) scene

differences, weather changes, sensor performance, and other issues resulting in varied UAV image quality, which hinders high-precision and highly robust object detection; (2) the complexity of the shooting scene and the uneven distribution of objects leading to the presence of noise and fuzzy boundaries on the features extracted using deep learning, which interferes with the performance; (3) different sizes and scales of objects in the images obtained by UAV affecting the accuracy of object detection; and (4) the domain gap between natural scene images and UAV images reducing the accuracy of pre-training models based on natural scene datasets when migrating to downstream UAV image object detection tasks.

To address the above-mentioned challenges, this study constructs a high-quality object detection method for UAV images. The object features extracted from UAV images were fully extracted and the object detection performance in UAV images was enhanced by fusing a high-performance feature extraction backbone and an optimizer with high stability and generalizability in the baseline network and utilizing the aero-space remote sensing scene dataset associated with the UAV image domain to obtain a pretraining model. First, based on the design concept of MetaFormer [27], we constructed a global–local hybrid feature extraction backbone that combines CNN and ViT to fully extract global and local features for the baseline network. Second, AdaBelief [28], which uses an adaptable step size, was used to achieve faster convergence, higher accuracy, better stability, stronger generalizability, and a more reasonable dynamic adjustment of the learning rate. Finally, by employing a dataset which has a very small domain gap with UAV images and a lower data volume relative to mainstream natural scene datasets, a pretraining model was developed based on the masked image modeling (MIM) method. Moreover, the object detection performance of the UAV images was further improved when migrating to downstream tasks. The results of the ablation study and experiments indicate that the proposed method can obtain results with comparable advantages to mainstream and state-of-the-art (SOTA) algorithms. The main contributions of this study are as follows:

(1) A high-quality UAV image object detection method was developed using the SOTA object detection method as the baseline network;

(2) A global–local hybrid backbone for feature extraction that combines the advantages of CNN and ViT was built for the baseline network;

(3) AdaBelief, an optimizer with better stability and generalizability, was used to achieve faster convergence and higher training accuracy by reasonably adjusting the learning rate;

(4) A pre-training model that better meets the downstream task requirements of UAV images was obtained using the MIM method and an aerospace image dataset with a much lower data volume than current mainstream natural scene datasets. This model further improved the object detection performance of UAV images.

The organization of the remainder of this article is as follows. In Section 2, several mainstream and SOTA object detection methods are reviewed. In Section 3, the proposed method is described. In Section 4, experiments comparing the proposed method with different methods are presented, and the limitations are discussed. Finally, in Section 5, the research is summarized, conclusions are drawn, and perspectives on future research are presented.

## 2. Related Work

### 2.1. Object Detection Methods

Many achievements have been made in the field of object detection; in particular, various types of intelligent algorithms have been rapidly developed, which include CNN-based, ViT-based, and CNN-ViT hybrid methods.

#### 2.1.1. CNN-Based Methods

Many CNN-based object detection methods have been proposed, such as the R-CNN series [4–9], YOLO series [10], SSD [11], RetinaNet [12], and FCOS [13]. As the network inference progresses during the extraction of feature maps using CNN-based methods,

object features and location information are gradually lost [29,30]. Therefore, researchers have developed feature pyramid networks (FPNs). Using multiscale methods, the effect of information loss can be effectively alleviated by extracting multiscale features. Based on this idea, FPN [31] combines bottom-up, top-down, and horizontal connections to comprehensively utilize pyramid features, thereby significantly improving the recognition accuracy of CNN-based object detection algorithms. Related algorithms using FPNs include Fast R-CNN [6], Masked R-CNN [7], and PANet [32]. To better utilize the FPN to obtain visual features with hierarchical differences and object features with different scales and saliency levels, researchers have conducted studies from multiple perspectives to improve object detection performance, for example, by enhancing the FPN [33], adding multipath detection calibration [34], and combining sequence and exception block (SE) modules [24].

With the development of improved architectures and better representation learning frameworks, modern ConvNets such as ConvNeXt [35] have achieved high performance in various tasks. Based on ConvNeXt, ConvNeXt V2 [36] included a completely convolutional shielded automatic encoder module and a new global response normalization (GRN) module, thereby realizing a combination of self-supervised learning.

To address scale changes in various objects within a large scene and detect objects with low pixels, FSoD-Net [37], a new single-stage detector for full-scale object detection, was proposed, which is cascaded by a multi-scale enhancement network (MSE-Net) and a scale-invariant regression layer. This network effectively improves the detection ability of small objects. In response to the problems of various object scales, complex backgrounds, and dense object distributions in various images, SME-Net [38] utilizes feature segmentation and merging modules to equipoise the features between different saliency objects, and then transfers the effective detailed feature information of large objects to the depth feature map, thus reducing the confusion of features between different scale objects.

### 2.1.2. ViT-Based Method

Although CNN-based methods for object detection have been widely employed in many fields, they cannot establish dependency relationships between features in cascaded structures because of the lack of perception of a global image. Instead, transformers, which were first proposed in the field of natural language processing, have a larger receptive field, more flexible weight setting, and global modeling ability in feature learning; thus, they can fully exploit contextual information and provide higher-quality features for downstream tasks. Therefore, researchers have migrated transformers to computer vision (CV) tasks, and high performances have been achieved in object detection, image classification, and image segmentation. Among these, ViT [14] is a pioneering application of transformers in CV. DETR [16] has provided an important foundation for the application of ViT in object detection tasks. Subsequently, many improved object detection algorithms based on DETR have been constructed, such as deformable DETR [17], DN-DETR [18], and DAB-DETR [19].

In particular, to solve the slow training convergence and unclear query meanings in DETR, DINO [39] combines a hybrid query selection algorithm with anchor initialization and a two-step forward-looking solution for box prediction based on DN DETR [18] and DAB-DETR [19]. It constructs an end-to-end object detector and uses comparative methods for denoising training. Compared to the previous DETR-like model, a significant improvement was observed in the performance and efficiency, and SOTA results were achieved on the COCO 2017 dataset.

Although ViT has unique advantages compared to CNNs, it has a secondary computational complexity problem closely related to the number of input images or tokens, resulting primarily in a large computational workload. Shortening the input sequence of the attention layer is an effective way to reduce the computational workload. To this end, Swin Transformer [20] and improved Swin Transformer V2 [40] included a hierarchical ViT constructed using sliding windows, introduced localization to capture local window attention, and utilized multiscale feature models, such as FPN, for dense prediction, which significantly reduced computational complexity while improving model performance.

However, the Swin Transformer still uses the object detection network of the CNN object detection model, resulting in a model that relies on a precise prior design, which poses great difficulties for adjusting parameters and training. In addition, the merging operation of adjacent patches by the Swin Transformer can lead to the loss of detailed local features [32]. Therefore, the CSwin Transformer [41] was developed to utilize cross-shaped windows to alleviate the shortcomings of the Swin Transformer, whereas the SepViT [42] was inspired by deep separable convolution and based on a separable visual converter. In the SepViT, information exchange is performed within and between windows through a deeply separable self-attention method, which can capture long-range visual dependencies in multiple windows.

Drawing on the principle of pyramid features, a progressive shrinking pyramid structure and spatial reduction attention mechanism were applied to the PVT model [21,22] to reduce the spatial dimension of the input sequence through the hierarchical attention of the image resolution, thereby reducing the computational complexity. To enable multiscale feature interaction capabilities in ViT, CrossFormer [43], which is based on local and global interactions, follows the feature pyramid structure in the PVT to obtain multiscale features, and it then utilizes a cross-scale embedding mechanism to achieve multiscale feature fusion. This process promotes the performance without remarkably improving the computational complexity relative to ResNet and PVT.

In a deformable attention transformer (DAT) [44], a deformable self-attention mechanism was constructed, selects the positions of key and value pairs based on data dependency. In CG-Net [45], a transformer was used to enhance the connection between channels, adaptively control the calibration weights of each channel, and aggregate and represent each channel. Wang et al. [46] proposed a ViT model for remote sensing tasks using a new rotated varied-size window attention instead of the full attention in the vanilla ViT, and extracted rich contextual information from different generated windows to learn better representations. To achieve multiscale object detection, ABNet [47] was designed to utilize enhanced and efficient channel attention to enhance the backbone's feature representation ability. Subsequently, by combining multiscale information from different channels and spatial positions, an adaptive feature pyramid model was used to extract more distinctive features. UNetFormer [48] used an efficient global–local attention module based on UNet to process global and local extracted information within the decoder, and it achieved good results.

### 2.1.3. Hybrid Methods

Each CNN- and ViT-based method has its own advantages and disadvantages in object detection tasks. CNN can retain local information with certain limitations in capturing long-range dependencies, whereas ViT can model global information but lacks the capability of retaining local feature details. Therefore, combining CNN and ViT to obtain high-quality object detection accuracy to improve the performance of object detection in UAV images is a research hotspot and is very important.

In terms of the serial integration of CNN and ViT, CMT was the first method [49] to utilize a CNN to obtain local features and apply the attention module to construct the global dependencies of local features. Residual links were used in multiple parts of CMT to preserve local features and to achieve the combination of local and global features. Yu et al. [27] performed research and experiments on various current ViTs and improvement methods, and concluded that the general architecture of ViT rather than specific token mixers results in significant performance. Based on the research results, they proposed MetaFormer [27], a general architecture abstraction from ViT without specific token mixers, and demonstrated its effectiveness through experiments.

In terms of the parallel integration of a CNN and ViT, GLSAN [23] was constructed to achieve excellent object detection results in UAV images by combining a global–local object detection baseline, a simple and efficient self-adaptive region selection algorithm, and a local super-resolution module. In DeiT [50], knowledge distillation was performed,

and the features learned via CNN were integrated into the ViT training process, which achieved the fusion of the two features. In Mobile-Former [51], a parallel structure was designed that bridges MobileNet and attention mechanisms, thus making feature fusion more effective and providing faster inference speeds when processing high-resolution images. In MPViT [52], multiple parallel encoders and convolutions are used to achieve global and local feature sharing for SOTA performance. In Conformer [53], parallel CNN and ViT branches were designed, and bridging modules were used to achieve feature fusion.

In addition to the above methods, to fully utilize the correlation between CNN and ViT, a non-local block was embedded in the backbone of FRPNet [54] to achieve the correlation of different regions in the image. By fusing adjacent lower-level fine-grained features and combining them with the proposed feature backflow pyramid structure, high-quality results of feature representations were generated for each scale, which improved the detection capabilities for multi-scale and multi-category objects. In ACmix [55], a thorough analysis of the similarity of feature extraction mechanisms between CNN and ViT was performed, and this method accomplished both CNN and ViT through shared feature mapping parameters and achieved good results on the COCO 2017 dataset. For MCCNet [56], a multi-content complementary module was constructed that connects the encoder and decoder to achieve salient object detection in multi-scale images.

### 2.2. Masked Modeling Methods

Supervised pretraining using massive natural scene-labeled datasets such as ImageNet [57] and COCO [58] is a mainstream approach for ensuring good object detection performance [59]. However, training object detection networks from scratch on mainstream large-scale datasets takes a long time and requires a higher-performance training environment. Moreover, a domain gap exists between the images of natural and UAV aerial photography scenes, which indicates that the pre-training model using natural scene images is inappropriate for the downstream tasks of UAV image object detection. To this end, researchers began using large-scale datasets in specific fields for pre-training and migrated to downstream tasks. For example, to achieve better object detection performance in aerospace images, Wang et al. [60] conducted empirical research on pre-training for aerospace images, and used the classification dataset MillionAID to train different networks from scratch. These authors obtained a series of pre-training models for remote sensing and achieved good results in downstream tasks.

However, the massive labeled dataset required for supervised pre-training remains difficult to obtain. Therefore, the focus of visual feature learning has shifted from supervised to self-supervised pre-training. Currently, self-supervised learning methods based on MIM, such as masked autoencoder (MAE) [61] and simple framework for masked image modeling (SIMMIM) [62], have been proposed and shown to be effective for downstream tasks [46].

MAE [61] was used to develop an extensible self-supervised learning machine for CV by reconstructing the randomly masked parts in the input images. Moreover, a well-summarized high-capacity model can be learned, which further increases the potential of ViT through feature pre-training and a multiscale hybrid of CNN and ViT. As a neural network pre-training framework, MAE has shown excellent performance in visual recognition, and thus has been applied in object detection in remote sensing images [63]. However, because of the complexity of MAE, overfitting problems may occur on small datasets. Therefore, by introducing methods such as position prediction and comparative learning tasks, combined with the localization and invariance features of MAE, methods such as FCMAE [36] and MCMAE [64] can significantly improve the performance of MAE.

Compared to MAE, SIMMIM [62] is a simple framework which aligns well with the nature of visual signals, and can learn better representations by using a moderately masked patch size to randomly mask input images, directly predict the missing pixels in the input image using regression, and combine an extremely lightweight prediction head to obtain strong representation learning abilities through the above simple design, which can achieve a similar or slightly better transferring performance with a remarkable speedup in pre-

training than that of heavier prediction heads. In addition, SIMMIM inputs the masked part into the feature extraction module, which is suitable for models that maintain the image structure. Overall, SIMMIM can effectively implement representation learning while maintaining a very simple structure.

## 3. Proposed Method

### 3.1. Overall Framework

As shown in Figure 1, the main process of the proposed method can be divided into two steps: pre-training and fine-tuning.

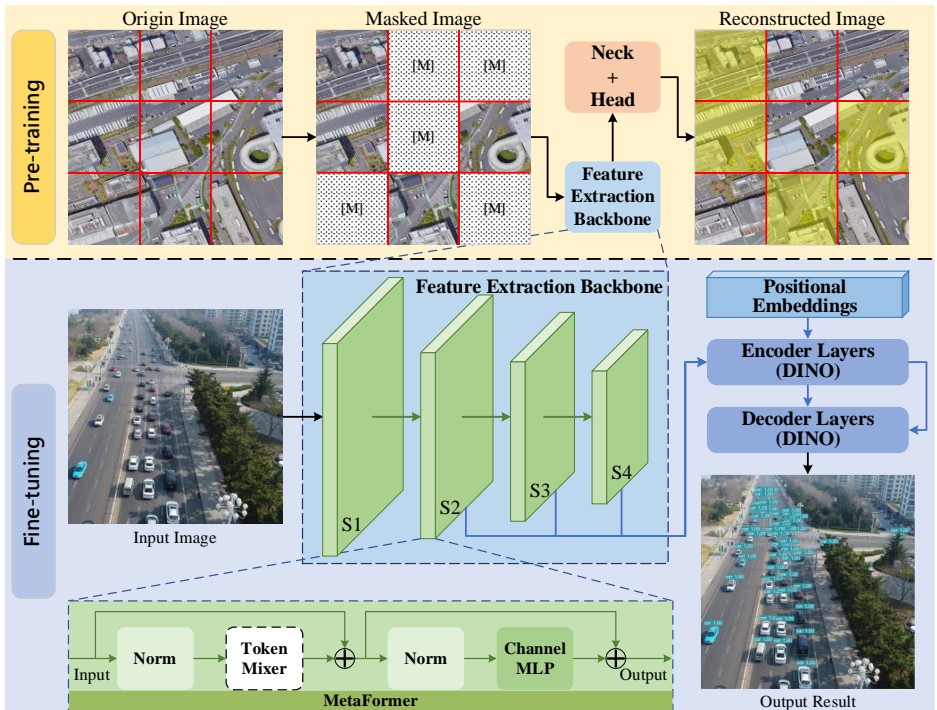

**Figure 1.** Pipeline of the proposed method for UAV images. In the reconstructed image of the pre-training step, the yellow part represents the unmasked content of the image. The features extracted from the feature extraction backbone are fed into the encoder and decoder layers for feature enhancement and result computation.

(1) In the pre-training step, a self-supervised training method based on SIMMIM was adopted. With the help of the significant similarity between the aerospace remote sensing images and UAV images, an aerospace remote sensing dataset, which has a much lower volume than the mainstream natural scene dataset, was applied to obtain the pre-training model of the proposed feature extraction backbone; thus, the pre-training model can better adapt to object detection in UAV images.

(2) In the fine-tuning step, DINO, which utilizes denoising methods to achieve high-precision object-detection accuracy, was used as the baseline network. In this study, we designed a backbone based on MetaFormer, as shown in Figure 1, and embedded a token mixer suitable for masked image training by combining the CNN and ViT to extract features, which will be transmitted to the encoder layers. The feature extraction backbone has four stages: S1, S2, S3, and S4, with each containing one downsampling module that was not presented and several MetaFormer blocks, introduced in Section 3.2 in detail. Notably, only the extraction features of stages of S2, S3, and S4 in the backbone are input to the encoder layers, which can further reduce computational complexity while maintaining a high performance. After that, the model used the optimizer AdaBelief, which has better stability and generalizability, to further improve network performance.

### 3.2. Global–Local Hybrid Feature Extraction Backbone

Relevant research shows that the excellent performance of ViTs is mainly attributed to the MetaFormer [27] structure. In addition, MetaFormer can further improve the performance by using different token mixers, especially mixed with CNN and ViT, which can fully utilize the advantages of convolutions and transformers to achieve better results. Therefore, to extract features efficiently, this study designed a global–local hybrid feature extraction backbone based on MetaFormer, as shown in Figure 2, and the process is as follows:

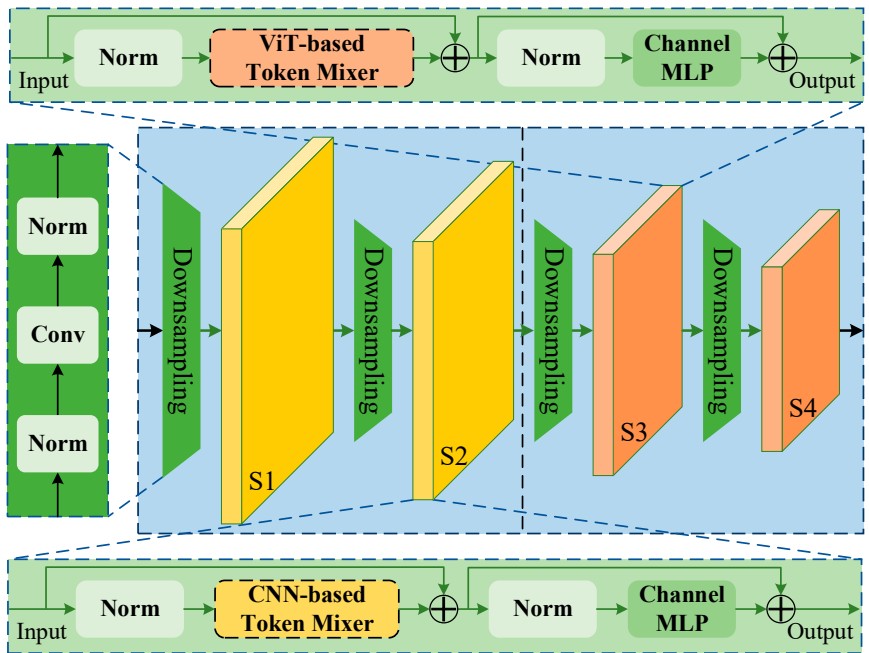

**Figure 2.** Architecture of the proposed feature extraction backbone. The backbone draws on the design philosophy of the MetaFormer and accomplishes global and local feature extraction hybridization. The backbone consists of four stages (S1, S2, S3, and S4), and each stage contains multiple MetaFormer blocks. The MetaFormer blocks in the first two stages (S1 and S2) contain the CNN-based token mixer, and the MetaFormer blocks in the last two stages (S3 and S4) contain the ViT-based token mixer. The downsampling module is employed to reduce the size of the input image or the feature from the pre-stage.

First, input data *I* are fed through the downsampling module to obtain the following features:

$$X = DownSampling(I) \tag{1}$$

where $X \in \mathbb{R}^{N \times C}$ represents the embedded tokens, and *N* and *C* represent the sequence length and dimension obtained through downsampling, respectively.

Subsequently, the embedding token *X* is input into the following stage. Each stage contains several MetaFormer blocks, and each MetaFormer block contains two residual modules. The first module is primarily used to extract and mix tokens and establish relationships between each token, which can be formulated as

$$X' = TokenMixer(Norm(X)) + X \tag{2}$$

where *Norm* represents standardization, which corresponds to layer normalization (LN) in this article, and *TokenMixer* is a module for mixing tokens and is mainly used to propagate token information. There are two types of token mixers: CNN-based and ViT-based token mixers, which are introduced in detail in Sections 3.2.1 and 3.2.2. The second residual

network contains a channel multilayer perceptron (MLP) with nonlinear activation, which can be represented as

$$X'' = \sigma(Norm(X')W_1)W_2 + X' \tag{3}$$

where $W_1 \in \mathbb{R}^{C \times rC}$ and $W_2 \in \mathbb{R}^{rC \times C}$ are leachable parameters and $\sigma(\cdot)$ represents the nonlinear activation function.

### 3.2.1. CNN-Based Token Mixer

Based on the standard ConvNet module, the ConvNeXt [35] module showed better performance in terms of accuracy, scalability, and robustness than models such as the Swin Transformer by utilizing different designs that have been validated by various methods over the past decade. More importantly, ConvNeXt maintains the efficiency of ConvNet, does not require complex modules such as shift window attention and relative position encoding, and is very easy to implement during training and testing. However, ConvNeXt is primarily aimed at supervised learning tasks, and the expected results cannot be obtained directly using ConvNeXt for unsupervised tasks [36]. Therefore, based on ConvNeXt, a new global response normalization (GRN) module was used in ConvNeXt v2 [36] to improve the feature competition between channels. The feature competition is enhanced when GRN is combined with the masked autoencoder method, which can achieve better performance. Given its excellent characteristics, the CNN-based token mixer was based on ConvNeXt v2, as shown in Figure 3.

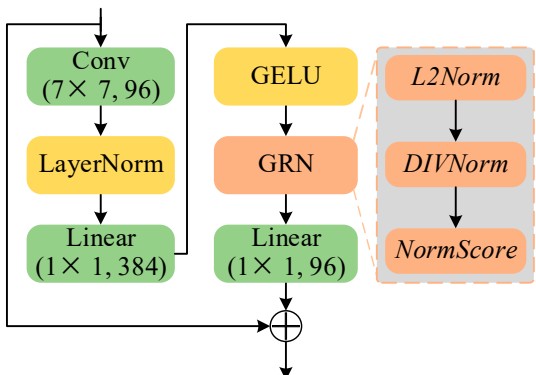

**Figure 3.** Architecture of the CNN-based token mixer for global feature extraction.

GRN can increase the contrast and selectivity of channels and is the main reason why ConvNeXt v2 can effectively utilize MAE for pre-training. In Figure 3, when spatial feature maps $X \in R^{H \times W \times C}$ are input into GRN, a feature aggregation layer implemented using the L2-norm in the *L2Norm* module had to be crossed to obtain the aggregated features:

$$X_{L2} = G(X) = L2 - norm(X) \tag{4}$$

where $X_{L2} = \{\|X_1\|, \|X_2\|, \ldots, \|X_C\|\}$ and $\|X_i\|$ represent the scalars of the statistics of the $i$-th aggregated channel. In GRN, *DIVNorm* uses the response normalization to calculate the relative importance score of aggregated features under a certain channel compared to features under other channels:

$$X_N^i = N(\|X_i\|) = \frac{\|X_i\|}{\sum_{j=1,2,\ldots,C}\|X_j\|} \tag{5}$$

Finally, the scores obtained from the *NormScore* module were used to calibrate the original inputs.

$$X_i = X_i * X_N^i \tag{6}$$

Meanwhile, to facilitate the optimization of GRN, the final GRN structure is

$$X_i = \gamma * X_i * X_N^i + \beta + X_i \tag{7}$$

where $\gamma$ and $\beta$ are two leachable parameters that are initialized with 0.

### 3.2.2. ViT-Based Token Mixer

ViT requires a large computational workload to achieve excellent performance. To address this problem, SepViT [42] combines depthwise and pointwise self-attention to construct a deep separable self-attention module. Local features within each window are obtained using depthwise self-attention, global attention is obtained between different windows using pointwise self-attention, and depthwise and pointwise self-attention are combined to capture local and global dependencies. To add bias to depthwise self-attention, this study draws on Swin Transformer [20] and DAT [44] and adopts relative position bias $B_{relative}$ to encode the relative positions between each query and key, thereby enhancing the spatial information of the attention mechanism. The ViT-based token mixer constructed in this study is illustrated in Figure 4.

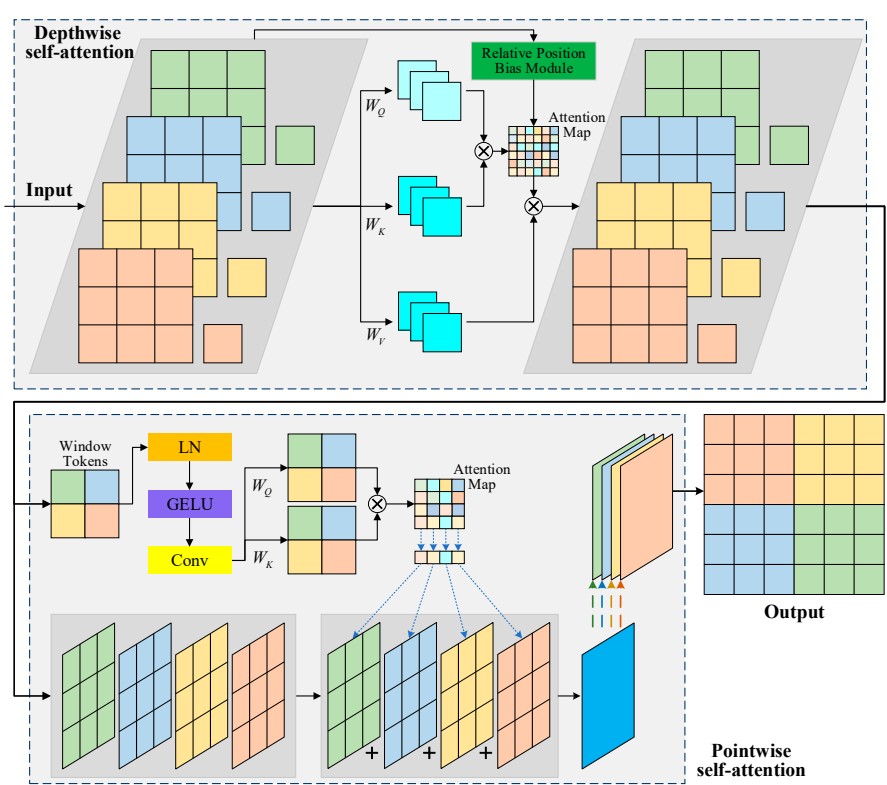

**Figure 4.** Architecture of the ViT-based token mixer for local feature extraction, which is an improved SepViT with relative position bias. Four windows are obtained when input data are divided in depthwise self-attention.

In depthwise self-attention, a series of windows are obtained after input data are divided (as shown in Figure 4). For each window, a window token is created that can achieve a global representation of the window, thereby modeling the relationship for attention in the subsequent processing.

Then, the operations of depthwise self-attention are carried out on each window and the corresponding window token to achieve the fusion of spatial information within each channel. The process can be expressed as follows.

$$Q = x \cdot W_Q, K = x \cdot W_K, V = x \cdot W_v \tag{8}$$

$$\widetilde{z} = QK^T / \sqrt{d} + B_{relative} \tag{9}$$

$$DWA(x) = \sigma(\widetilde{z})V \tag{10}$$

where $x$ represents the feature tokens; $W_Q$, $W_K$, and $W_V$ represent the linear layers for obtaining the query, key, and value, respectively; and $\sigma$ is the softmax function. After obtaining $\widetilde{z}$, a learnable relative position offset $B_{relative}(q)$ is added to $\widetilde{z}$, and then $DWA(x)$ can be obtained using the self-attention mechanism calculation method.

For the window and corresponding window token obtained by depthwise self-attention, due to the changes in their shape, this article constructs a reference window that is one size larger than the input feature map, as shown in Figure 5. A relative position encoding $B_{relative} \in \mathbb{R}^{(H \times W+1) \times (H \times W+1)}$ was obtained, where $H$ and $W$ represent the height and width of the window, respectively. The relative positions of a feature map on different coordinate systems are within the ranges $[-H, H]$ and $[-W, W]$. This project constructs a small-scale offset matrix $\hat{B} \in \mathbb{R}^{(2H-1) \times (2W-1)}$, calculates the relative displacement within the normalized range $[-1, +1]$, and interpolates $B_{relative}$ from $\hat{B} \in \mathbb{R}^{(2H-1) \times (2W-1)}$ to obtain all possible offset values. In addition, the relative position deviation learned in pre-training can also be initialized by bicubic interpolation to create a fine-tuning model with different window sizes.

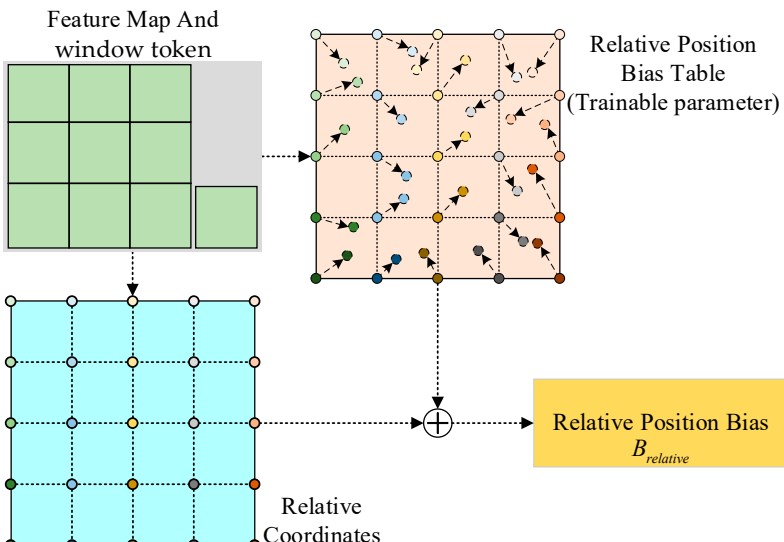

**Figure 5.** Relative position bias module.

### 3.3. Efficient Optimizer for Training

The optimizers used in most models are stochastic gradient descent (SGD) with acceleration strategies or adaptive AdamW. AdamW is faster than SGD but has poor generalizability. However, AdamW is widely used in complex networks owing to its stability.

In this study, the proposed method is relatively complex, and to obtain faster, better, and more stable optimization results, a more efficient optimizer is required. AdaBelief [28] was used as the optimizer in this study because of its fast convergence, which is similar to that of AdamW, and its better generalizability and training stability, which are similar to those of SGD.

AdaBelief is a modification based on Adam in which the step size is adjusted based on the current gradient direction. The prediction of the next time step is represented by the exponential moving average of the noise gradient. If the current gradient differs significantly from the prediction, then the current gradient direction is not trusted, and only a small step is taken; otherwise, the current gradient is trusted and a large step is taken. Many experimental results have shown that AdaBelief has faster convergence and

higher precision in image classification and language modeling, which indicate that it shows better performance.

Using $f$ as the optimization objective function, $\theta_t$ as the $t$-th iteration model parameter, $B_t$ as the training parameter, and $\eta$ as the learning rate, the optimization steps of AdaBelief are as follows:

$$g_t = \nabla_{\theta_{t-1}} f(B_t, \theta_{t-1}) \tag{11}$$

$$\theta_t = \theta_{t-1} - \eta g_t \tag{12}$$

$$M_t = (1 - \beta_1)g_t + \beta_1 M_{t-1} \tag{13}$$

$$s_t = (1 - \beta_2)(g_t - M_t)^2 + \beta_2 s_{t-1} \tag{14}$$

where $M_t$ is the momentum direction, and $\beta_1$ and $\beta_2$ are hyperparameters. The optimization process can be expressed as

$$\theta_t = \theta_{t-1} - \alpha \frac{M_t}{\sqrt{s_t} + \varepsilon} \tag{15}$$

where $\alpha$ represents the learning rate, and $\varepsilon$ is used to prevent the parameters from dividing by 0.

### 3.4. Masked-Based Pretraining Model

Currently, the backbones of various object detection methods are trained on natural scene datasets and migrate to downstream tasks. Owing to the domain gap between natural and UAV aerial photography scenes, existing pre-training models cannot be well applied to the object detection of the UAV image. Therefore, directly using existing pre-training models affects the accuracy improvement of UAV image object detection. In addition, existing natural scene datasets are too large, and require enormous computational resources during the pre-training process, which hinders academic research and practical implementation.

In response to the proposed overall framework and hybrid backbone, this study utilized publicly available aerospace remote sensing datasets with much less data volume than natural scene datasets (such as ImageNet-1k), which are similar to UAV images, to achieve better support for the object detection of UAV images. The pre-training process is implemented based on the SIMMIM model, as shown in Figure 6.

Compared to MAE, SIMMIM has a lightweight decoder and receives structured inputs, making it more suitable for extracting features using complex backbones containing ViTs. Additionally, during the model prediction process, SIMMIM directly predicts the pixels contained in the masked area through regression. However, because many currently designed backbones use down-sampled feature maps, SIMMIM maps each feature vector back to its original size and assigns this vector to predict the corresponding pixels to predict the original image data at full resolution.

In this study, the input image was first segmented according to a given patch size, and the divided image was then masked using a random mask strategy according to the mask rate. The masked image was used as the input for the subsequent process. The masked input data were then input into an encoder structure containing the proposed backbone, which extracts potential features for the masked images to predict the raw information of the masked area. In this study, the learned encoder structure was transferred to the model structure.

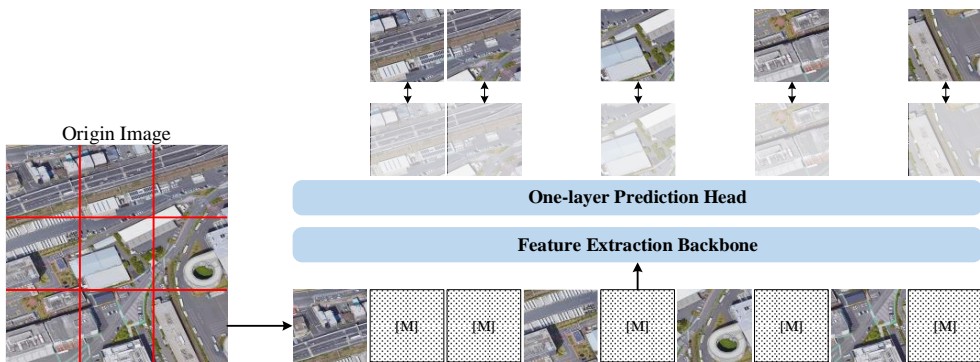

**Figure 6.** Pre-training model training process based on SIMMIM.

## 4. Experiments and Results

### 4.1. Experiments Setting

#### 4.1.1. Pre-Training Datasets

FAIR1M [65]: This dataset is a fine-grained high-resolution dataset of remote sensing scenes suitable for object detection with over 40,000 images and 1 million instances. The resolutions of images in FAIR1M range from 0.3 to 0.8 m and are distributed across multiple countries and regions in Asia, America, and Europe. All images in FAIR1M were annotated using directed bounding boxes involving 5 categories and 37 subcategories. A comparison with the current mainstream dataset used for pre-training is shown in Table 1.

**Table 1.** Information on the different datasets used for pre-training.

| Datasets | Images (k) | Instances (M) | Categories | Volume (GB) |
|---|---|---|---|---|
| COCO 2017 [58] | 328 | 2.5 | 91 | 25 |
| ImageNet-1k [57] | 1430 | 1.28 | 1000 | 144 |
| Objects365 [66] | 630 | 29 | 365 | 48 |
| DOTA v2.0 [67] | 11.3 | 1.79 | 18 | 138 |
| FAIR1M | 42.8 | 1.02 | 37 | 43 |

As shown in Table 1, the FAIR1M images have fewer images and categories than the COCO 2017, ImageNet-1k, and Objects365 datasets; however, the number of instances is relatively large. Moreover, the storage space required for FAIR1M is relatively low. Therefore, FAIR1M has high data quality and instance density. Compared to DOTA v2.0, FAIR1M has a slightly lower instance density but only occupies 31% of the storage space; moreover, it includes richer fine-grained categories, making it suitable for conducting model pretraining tasks and transferring to downstream tasks. More importantly, compared to the mainstream datasets used for pre-training, such as COCO 2017 and ImageNet-1k, the FAIR1M dataset mainly consists of remote sensing images. The acquisition method and environment of the image in FAIR1M dataset are similar to UAV images, and the resolution is also relatively similar, which indicates that the domain gap between FAIR1M and UAV images is relatively smaller.

#### 4.1.2. Fine-Tuning Datasets

The fine-tuning datasets used for object detection tasks in this study included the AFO and UAVDT.

UAVDT [68]: This dataset is based on vehicle traffic content captured by UAVs, contains approximately 8000 frames of images selected from 10 h of raw video, and can be used in the tasks of object detection, single-object tracking, and multi-object tracking. In the UAVDT dataset, 75.0%, 15.0%, and 10.0% of the UAV images were used for training, testing, and validation, respectively.

AFO [69]: This dataset was extracted from over 50 UAV videos for deep learning with resolutions ranging from $1280 \times 720$ to $3840 \times 2160$, and it includes 3647 images with 39,991 manually labeled individuals and floating objects on water, many of which are small-scale objects that are difficult to detect. In this dataset, 76.6%, 14.1%, and 9.3% of the images were used for training, testing, and validation, respectively.

### 4.1.3. Implementation Details

The processes of pre-training and fine-tuning in this study used the same training workbench running on an NVIDIA GeForce RTX 3090 GPU.

The process of obtaining a pre-trained model of the proposed backbone in this study was based on SIMMIM using the aerospace remote sensing dataset FAIR1M. The relevant hyperparameters in SIMMIM are shown in Table 2.

**Table 2.** Relevant hyperparameters in SIMMIM during pre-training.

| Item | Parameter | Value |
|---|---|---|
| | Epochs | 300 |
| | Batch size | 12 |
| | Optimizer | AdamW |
| | Learning rate | 0.0008 |
| Pre-training hyperparameter | Weight decay | 0.05 |
| | Random crop size | 448 |
| | Random crop scale | (0.67, 1.0) |
| | Random crop ratio | (0.75, 1.33) |
| | Random flip | 0.5 |
| | Input size | 448 |
| Mask generator | Mask patch size | 32 |
| | Model patch size | 1 |
| | Mask ratio | 0.6 |

All backbones involved in the comparison used the officially released models to initialize the parameters, and these models are obtained based on the ImageNet-1K dataset. Except for ResNet-50, ResNet-101, and ResNeXt-101, which had 200 training epochs, all other backbones had 300 training epochs. Some 50 epochs were employed in the fine-tuning stage. The images in UAVDT and AFO were resized to 448, and the random flip was 0.5. The optimizers used in the proposed and comparative methods were AdaBelief and AdamW, respectively. All optimizers used a learning rate of 0.0001 and a weight decay of 0.0001.

### 4.1.4. Evaluation Metrics

The metric used to evaluate the performance of the proposed and comparative method was mean average accuracy (mAP) of the dataset [37,45], which was obtained as follows:

$$AP_i = \int_0^1 P_i(R)dR \tag{16}$$

$$mAP = \frac{1}{C}\sum_{i=1}^{C} AP_i \tag{17}$$

where $R$, $P_i(\cdot)$, and $AP_i$ represent the recall rate accuracy rate, and average accuracy of category $i$, respectively, $C$ represents the number of categories, and $i = (1, 2, \ldots, C)$. The precision recall (PR) curve was also used to indicate the different performances of each method. For a specific category, the PR curve was obtained by calculating the precision and recall values.

$$Precision = \frac{TP}{TP + FP} \tag{18}$$

$$Recall = \frac{TP}{TP + FN} \tag{19}$$

where *TP* represents a true positive, which means that the intersection-over-union (IoU) between the obtained bounding box and ground truth (GT) is greater than a given threshold; otherwise, a false positive (*FP*) was obtained. When an object in the GT region does not have a corresponding bounding box in the detection results, the GT is treated as a false negative (*FN*).

### 4.2. Ablation Study

The most significant feature of DINO is its use of comparative denoising training methods to achieve high-performance end-to-end object recognition. However, the performance of DINO is poor when applied to detect objects in UAV images. Therefore, DINO was used in this study as the baseline to construct a method for obtaining high-quality object detection results for UAV images. DINO was implemented in MMDetection, an object detection toolbox and benchmark, and the hyperparameters were consistent with the vanilla DINO [39], where the number of queries was 900, a six-layer transformer encoder in the encoder layers and a six-layer transformer decoder in the decoder layers were used, and the hidden feature dimension was 256. Based on the baseline network, an ablation study was conducted using a backbone, an optimizer, and a dataset.

Global–Local Hybrid Feature Extraction Backbone: Based on MetaFormer, this study used a CNN-based token mixer in the first two stages and a ViT-based token mixer in the latter two stages. By combining the characteristics of CNN and ViT, a global–local hybrid feature extraction backbone was constructed, which contained stages S1, S2, S3, and S4. The block depths were 6, 12, 14, and 2, and the dimensions were 64, 128, 320, and 512. Although the network has four stages, the proposed method in this study only uses features from S2 to S4 for transmission to the encoder layers (as shown in Figure 1).

Efficient Optimizer: This study used AdaBelief as the optimizer to modify the step size based on different gradient conditions obtained and to achieve faster convergence, higher precision, and better performance.

Based on the UAVDT and AFO datasets, the ablation study was conducted and the results are listed in Table 3. The vanilla backbone for DINO was ResNet-101, and the default optimizer was AdamW. The proposed method adopts the proposed global–local hybrid feature extraction backbone and conducts pre-training on the FAIR1M dataset. The optimizers used were AdamW and AdaBelief.

**Table 3.** Results of ablation study based on the UAVDT and AFO datasets. The proposed backbone is the global–local hybrid feature extraction backbone.

| Method | Dataset | Backbone | Optimizer | Params (M) | FLOPs (G) | mAP$_{50}$(%) |
|--------|---------|----------|-----------|------------|-----------|---------------|
| DINO | UAVDT | ResNet-101 | AdamW | 66.21 | 79.72 | 82.3 |
| | | Proposed Backbone | AdamW | 57.69 | 80.47 | 82.2 |
| | | | AdaBelief | 57.68 | 80.47 | 82.2 |
| | AFO | ResNet-101 | AdamW | 66.21 | 79.72 | 56.3 |
| | | Proposed Backbone | AdamW | 57.69 | 80.47 | 56.7 |
| | | | AdaBelief | 57.68 | 80.47 | 57.6 |

As shown in Table 3, the proposed method significantly reduced the number of parameters by more than 10%. Simultaneously, the proposed method achieved the same or even better performance than the comparison methods with the AdamW optimizer, but did not significantly improve the model complexity. When different optimizers are used, the proposed method in this study can effectively enhance the object detection accuracy, particularly for the AFO dataset.

### 4.3. Experiment Results

In this section, the performance comparison results between the proposed method and mainstream SOTA methods based on the UAVDT and AFO datasets are present.

#### 4.3.1. Experimental Results for UAVDT

The input images of UAVDT were resized to $448 \times 448$ without maintaining the scale. A detailed comparison was conducted, and the evaluation metrics included the object detection precision, model parameters, and computational complexity, as shown in Table 4.

**Table 4.** Performance comparison for object detection for the UAVDT dataset. All models have a fine-tuning training epoch of 50. The unit of model parameter quantity is million, and the model complexity is represented by floating point operations per second (FLOPs). The values marked with red, green, and blue represent the three best-performing models.

| Methods | Backbone | Params (M) | FLOPs (G) | mAP (%) | AP50 (%) | AP75 (%) | APS (%) | APM (%) | APL (%) |
|---|---|---|---|---|---|---|---|---|---|
| Mask R-CNN [7] | Swin-S [20] | 66.07 | 55.46 | 65.2 | 76.7 | 75.6 | 38.6 | 85.7 | 91.3 |
| | ActiveMLP-B [70] | 69.21 | 56.32 | 63.4 | 77.3 | 73.9 | 36.8 | 83.4 | 85.8 |
| | CAFormer-M36 [27] | 69.75 | 72.24 | 66.4 | 77.8 | 76.0 | 41.7 | 86.3 | 89.6 |
| | MViTv2-B [71] | 67.94 | 56.89 | 65.0 | 76.5 | 74.6 | 38.1 | 85.3 | 90.6 |
| | BiFormer-B [72] | 73.29 | 58.76 | 67.6 | 76.8 | 76.3 | 41.4 | **88.4** | 92.5 |
| Cascade R-CNN [8] | ResNet-50 [73] | 68.94 | **40.83** | 68.1 | 78.7 | 77.6 | 44.2 | 86.8 | **94.1** |
| | DAT-T [44] | 76.64 | 385.00 | 66.3 | 76.8 | 75.4 | 37.7 | **87.3** | 92.2 |
| | ConvNeXt-T [35] | 76.82 | 385.37 | 65.3 | 76.5 | 75.0 | 38.7 | 85.9 | **93.9** |
| | ConvNeXt V2-T [36] | **63.92** | 377.23 | 62.1 | 70.9 | 69.8 | 26.0 | **87.4** | 93.0 |
| | SMT-B [74] | 80.44 | 398.33 | 63.3 | 71.5 | 70.8 | 30.4 | **88.4** | **94.6** |
| HTC [75] | ResNet-50 [73] | 68.93 | **40.84** | 66.5 | 79.6 | 75.2 | 40.5 | 86.4 | 93.0 |
| FCOS [13] | ResNeXt-101 [76] | 89.63 | 85.22 | **79.3** | 95.3 | **90.6** | **67.0** | 86.8 | 92.6 |
| Deformable DETR [17] | ResNet-101 [73] | **58.76** | **54.38** | 74.5 | **97.1** | 88.9 | 65.4 | 80.9 | 86.4 |
| RetinaNet [22] | PVTv2-B3 [22] | 70.70 | 55.20 | 68.7 | 90.7 | 76.7 | 41.0 | 83.0 | 92.5 |
| DINO [39] | ResNet-101 [73] | 66.21 | 79.72 | **82.3** | **98.9** | **95.4** | **75.1** | 87.1 | 92.8 |
| Proposed Method | | **57.68** | 80.47 | **82.2** | **98.8** | **95.7** | **74.6** | 87.1 | 93.5 |

As shown in Table 4, for mAP, the proposed method achieved a very similar performance to that of the optimal model (Vanilla DINO) and reduced the number of parameters by 12.88%. Some metrics, such as AP75, were better than those of the optimal model. In addition, the proposed method has a more significant performance advantage while maintaining a minimum quantity of parameters. For example, the proposed method has slightly fewer parameters than Deformable DETR, but significantly improves the performance. In addition, the proposed method has achieved a high performance in terms of object detection of small size. However, the FLOPs of the proposed method increased significantly.

A comparison of the P–R curves between the proposed and comparative methods in different categories is shown in Figure 7. The results indicate that the proposed method achieved strong performance in different categories.

To intuitively demonstrate the effectiveness of the proposed method, partial object detection results from the UAVDT dataset were selected, and are shown in Figures 8 and 9. The findings show that the proposed method can obtain relatively high performances in low-visibility and poor imaging environments (for example, during nighttime), and has fewer object detection errors, especially for small objects.

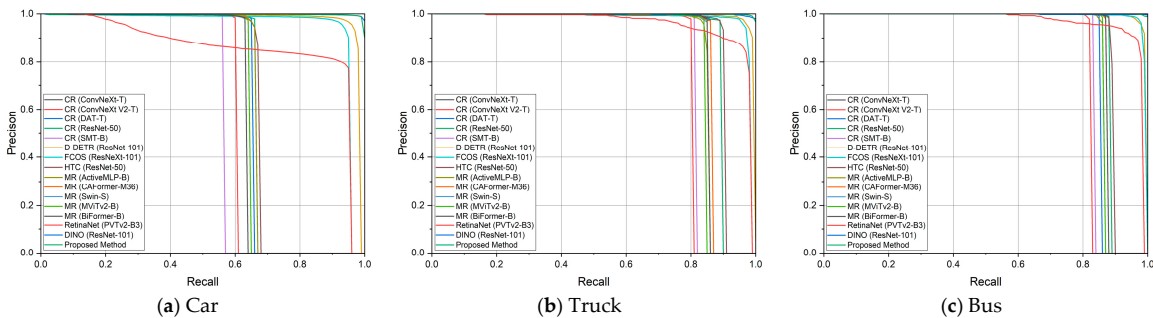

**Figure 7.** P–R curves of different categories obtained using different methods on the UAVDT dataset.

**Figure 8.** Visualization results of object detection for the first selected image in the UAVDT dataset. MR represents Mask R-CNN, CR represents Cascade R-CNN, and D-DETR represents Deformable DETR.

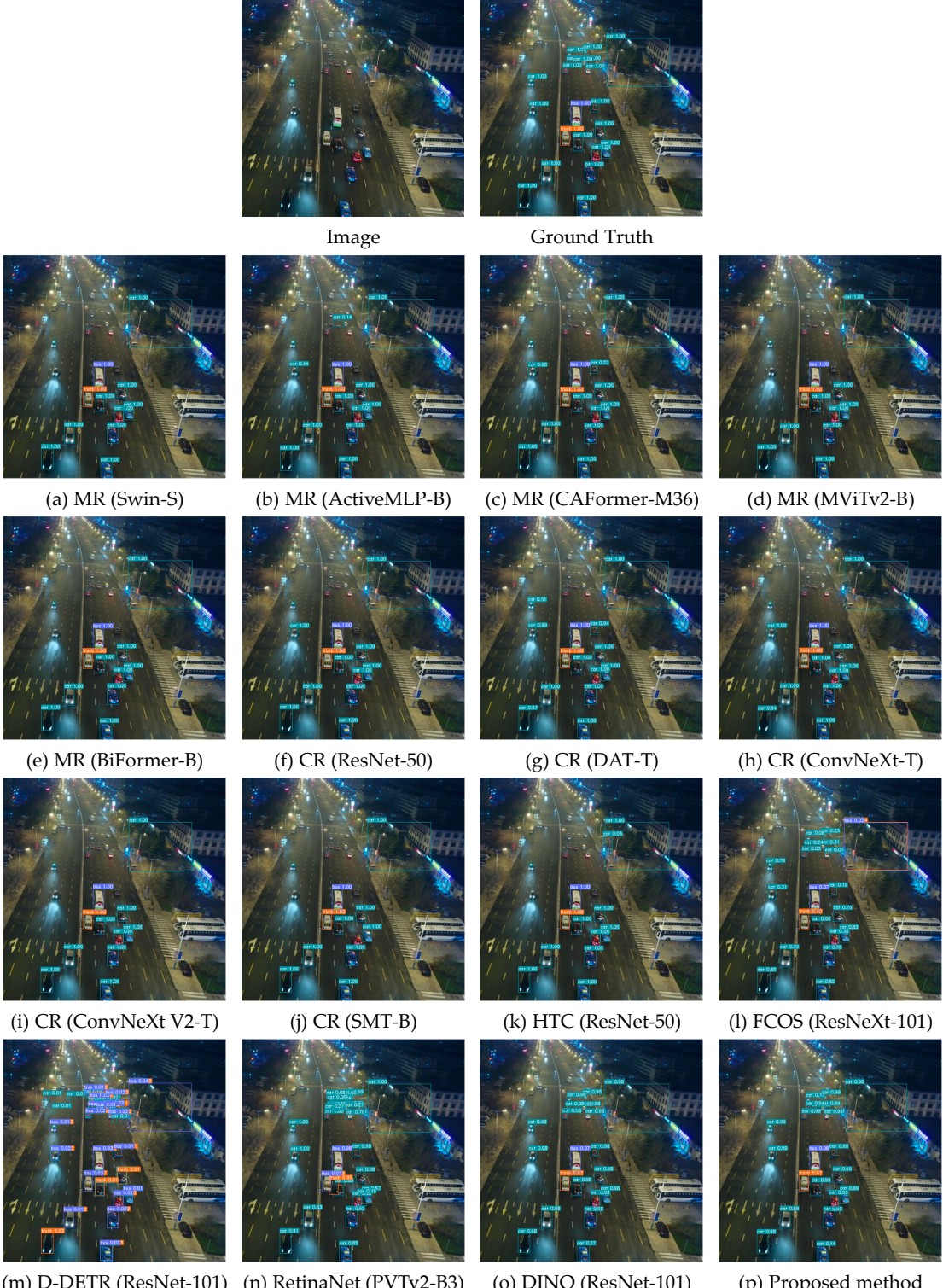

**Figure 9.** Visualization results of object detection for the second selected image in the UAVDT dataset. MR represents Mask R-CNN, CR represents Cascade R-CNN, and D-DETR represents Deformable DETR.

### 4.3.2. Experimental Results for AFO

The input images of AFO were reshaped as that of the UAVDT without maintaining the scale. A detailed comparison was conducted, and the evaluation metrics included object detection precision, model parameters, and computational complexity, as shown in Table 5.

**Table 5.** Performance comparison for object detection for the AFO dataset. All models have a fine-tuning training epoch of 50. The unit of model parameter quantity is million, and the model complexity is represented by FLOPs. The values marked with red, green, and blue represent the three best-performing ones.

| Methods | Backbone | Params (M) | FLOPs (G) | mAP (%) | AP50 (%) | AP75 (%) | APS (%) | APM (%) | APL (%) |
|---|---|---|---|---|---|---|---|---|---|
| Mask R-CNN [7] | Swin-S [20] | 66.07 | 55.46 | 47.4 | 66.2 | 55.3 | 0.0 | 37.8 | 66.6 |
| | ActiveMLP-B [70] | 69.21 | 56.32 | 43.3 | 62.3 | 50.9 | 0.4 | 28.0 | 61.8 |
| | CAFormer-M36 [27] | 69.75 | 72.24 | 41.6 | 62.9 | 46.3 | 0.3 | 31.9 | 59.0 |
| | MViTv2-B [71] | 67.94 | 56.89 | 43.0 | 63.5 | 50.0 | 0.1 | 33.6 | 61.6 |
| | BiFormer-B [72] | 73.29 | 58.76 | 48.6 | 67.1 | 56.7 | 0.0 | 36.3 | 68.0 |
| Cascade R-CNN [8] | ResNet-50 [73] | 68.94 | 40.83 | 50.0 | 67.9 | 59.2 | 1.5 | 36.8 | 69.4 |
| | DAT-T [44] | 76.64 | 385.00 | 46.8 | 66.7 | 55.3 | 0.3 | 35.5 | 65.2 |
| | ConvNeXt-T [35] | 76.82 | 385.37 | 51.7 | 71.3 | 61.4 | 0.6 | 40.3 | 71.0 |
| | ConvNeXt V2-T [36] | 63.92 | 377.23 | 38.9 | 57.2 | 45.0 | 0.0 | 24.8 | 56.7 |
| | SMT-B [74] | 80.44 | 398.33 | 48.4 | 63.7 | 57.9 | 0.0 | 28.3 | 69.6 |
| HTC [75] | ResNet-50 [73] | 68.93 | 40.84 | 49.0 | 68.2 | 56.4 | 0.5 | 35.4 | 68.7 |
| FCOS [13] | ResNeXt-101 [76] | 89.63 | 85.22 | 54.5 | 83.0 | 60.8 | 21.8 | 42.8 | 67.9 |
| Deformable DETR [17] | ResNet-101 [73] | 58.76 | 54.38 | 48.2 | 82.9 | 49.9 | 10.0 | 43.0 | 58.3 |
| RetinaNet [22] | PVTv2-B3 [22] | 70.70 | 55.20 | 46.2 | 76.8 | 48.5 | 13.6 | 32.0 | 59.3 |
| DINO [39] | ResNet-101 [73] | 66.21 | 79.72 | 56.3 | 90.9 | 62.4 | 29.9 | 54.0 | 66.1 |
| Proposed Method | | 57.68 | 80.47 | 57.6 | 89.7 | 62.5 | 30.3 | 51.7 | 67.9 |

The proposed method has obvious advantages for object detection (Table 5). For mAP, the proposed method obtained optimal performance in terms of AP and AP75, and high performance in terms of AP 50. In addition, the proposed method showed a strong performance in the detection of objects of different sizes, especially for small objects, and was significantly better than other methods, proving its effectiveness for multi-scale objects.

A comparison of the P–R curves between the proposed and comparative methods in different categories is shown in Figure 10. The results indicate that the proposed method achieved high performance in different categories.

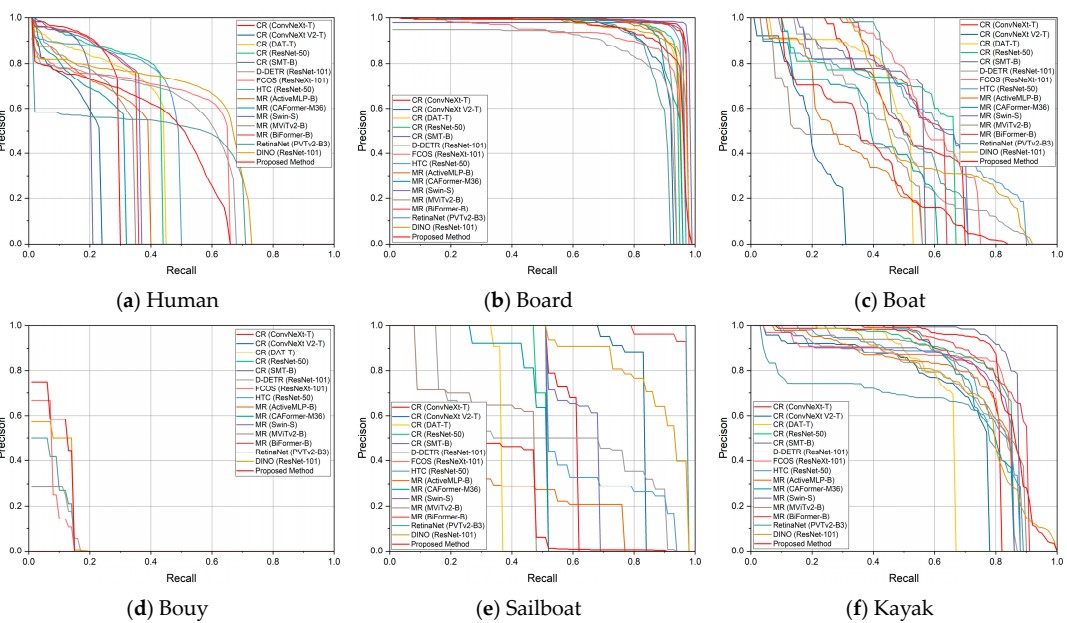

**Figure 10.** P–R curves of different categories obtained using different methods on the AFO dataset.

To intuitively demonstrate the better performance of the proposed method, the results of partial object detection for the AFO dataset were selected, and are shown in Figure 11. The proposed method achieves relatively good results in low-visibility and poor imaging environments (for example, at nighttime), and has fewer object detection errors, especially for small objects.

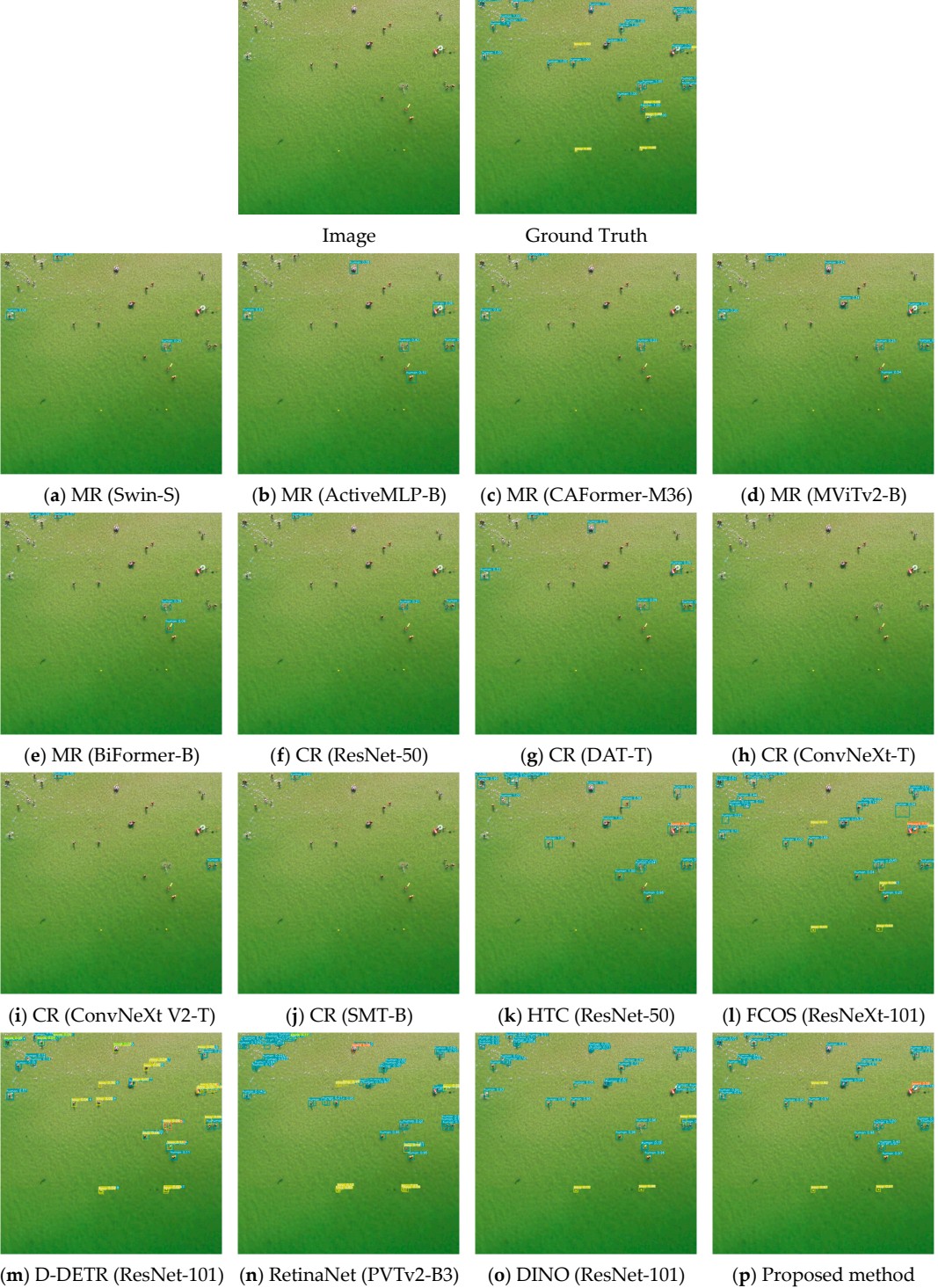

**Figure 11.** Visualization results of object detection for the selected image for the AFO dataset. MR represents Mask R-CNN, CR represents Cascade R-CNN, and D-DETR represents Deformable DETR.

### 4.4. Limitations

Based on the SOTA model DINO, this study presented a higher-performance global–local hybrid feature extraction backbone for UAV image object detection, utilized a pretraining method SIMMIM to construct a pretraining model, combined it with a more robust optimizer to achieve better performance compared to other SOTA models, and effectively reduced the number of parameters. Nevertheless, the computational complexity of the proposed method is still high, and the completion of convergence takes a long time. In addition, the proposed method has no significant advantage in some metrics, and requires further improvements. Finally, the image processing efficiency of the proposed method is lower than that of the other SOTA models, and requires further optimization.

## 5. Conclusions

In this study, a high-quality object-detection method for UAV images was developed, and DINO was used as the baseline. Based on MetaFormer, a CNN and ViT hybrid feature extraction backbone that can fully extract global and local feature information was constructed, and an optimizer with better stability and generalizability was adopted to obtain more efficient convergence and better performance during training. More importantly, the proposed method can achieve optimal and high performances of some metrics and reduce the number of parameters, as indicated in Tables 4 and 5. Simultaneously, to solve the problem of inappropriateness of pretrained models based on natural scenes for the object detection of UAV images, FAIR1M, an aerospace remote sensing scene dataset, which is much smaller than the natural scene dataset, was employed for the pretraining process based on the SIMMIM method. Then, it was migrated to the proposed method to fully extract features in UAV images to accomplish high-quality object detection, especially according to the AP75 metrics, to achieve optimal results for the UAVDT and AFO datasets.

The proposed method achieved better object detection accuracy compared to SOTA methods with fewer pretraining datasets and parameters, which can further promote the effective application of UAV images. However, the proposed method has shortcomings, as discussed in Section 4.4; therefore, the construction of high-precision and -throughput and lightweight models will be considered in subsequent studies. For this purpose, lighter network structures (e.g., RetinaNet), a more efficient backbone (e.g., MobileViT), and more reasonable training methods (e.g., knowledge distillation) will be adopted.

**Author Contributions:** Conceptualization, W.L. (Wanjie Lu) and C.L.; methodology, C.N. and W.L. (Wei Liu); validation, S.W.; formal analysis, S.W. and J.Y.; investigation, W.L. (Wanjie Lu); resources, W.L. (Wei Liu) and J.Y.; data curation, W.L. (Wei Liu) and J.Y.; writing—original draft preparation, W.L. (Wanjie Lu); writing—review and editing, C.L., C.N. and W.L. (Wei Liu); visualization, W.L. (Wanjie Lu); supervision, T.H.; project administration, W.L. (Wei Liu) and T.H.; funding acquisition, W.L. (Wanjie Lu). All authors have read and agreed to the published version of the manuscript.

**Funding:** This research was funded in part by the National Natural Science Foundation of China under Grant 42201472. We would like to express our sincere appreciation to the anonymous reviewers and editors for their constructive comments and suggestions.

**Data Availability Statement:** The UAVDT datasets were freely provided by Baidu, https://aistudio.baidu.com/aistudio/datasetdetail/106756 (accessed on 10 August 2023); the AFO datasets were freely provided by Kaggle, https://www.kaggle.com/datasets/jangsienicajzkowy/afo-aerial-dataset-of-floating-objects (accessed on 10 August 2023); The FAIR1M datasets were freely provided by Chinese Academy of Sciences, https://aistudio.baidu.com/aistudio/datasetdetail/78453 (accessed on 10 August 2023).

**Acknowledgments:** We gratefully thank the anonymous reviewers and editors for their constructive comments and suggestions. Simultaneously, we gratefully thank OpenMMLab for their object detection toolbox MMDetection.

**Conflicts of Interest:** The authors declare no conflict of interest.

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
