# Peer review of "High-Quality Object Detection Method for UAV Images Based on Improved DINO and Masked Image Modeling"

_remotesensing, doi:10.3390/rs15194740_

Round 1

Reviewer 1 Report

The manuscript named “High-quality Object Detection Method for UAV Images Based on Improved DINO and Masked Image Modeling” has been reviewed. This work attempted to combine the SIMMIM and a hybrid backbone based on CNN and ViT to improved UAV image object detection accuracy. In general, the paper is well-written and well-organized, the results seem reliable. However, there are still several problems that need to be improved.

My concern and the specific suggestions are as follows:

1. The motivation of using the hybrid backbone based on CNN and ViT, and why design this method are unclear.

2. In the experiments, I suggest the authors to add more comparisons with Transformer-based methods.

3. Some components in Figure 1 need to be briefly introduced, such as the Encoder Layer and Decoder Layers of DINO.

4. The entire article needs to be sorted out in order to correct any grammar errors that exist.

5. The similarity between FAIR1M and the data used in this article, such as UAVDT, needs to be explained in more detail.

6. A more detailed description of the comparison of different algorithms is needed.

7. The conclusion section should be supported by results from comparisons, and the subsequent studies needs to be slightly clarified.

8. Some pictures’ resolution are not high. Please make an improvement.

Author Response

  1. The motivation of using the hybrid backbone based on CNN and ViT, and why design this method are unclear.

Response: CNN- and ViT-based methods has their own advantages and disadvantages in object detection tasks. The CNN-based methods can consider the local features of the UAV images, but has certain limitations in capturing long-range dependencies. The ViT-based methods can provide long-distance feature dependencies. However, it lacks the capability of retaining local feature details. To improve the performance of object detection in UAV images, an effective way is to integrate global perception and local perception by combining CNN and ViT networks.

The above exploitation is added in the begin of section “2.1.3. Hybrid Methods”.

  1. In the experiments, I suggest the authors to add more comparisons with Transformer-based methods.

Response: Considering the Reviewer’s suggestion, we add several comparison Transformer-based methods, include SMT and BiFormer. The comparison results are shown in Table 4, Table 5, Figure 7, Figure 8, Figure 9, Figure 10, and Figure 11.

  1. Some components in Figure 1 need to be briefly introduced, such as the Encoder Layer and Decoder Layers of DINO.

Response: Considering the Reviewer’s suggestion, we add briefly introductions and explanations for the component in DINO.

  1. The entire article needs to be sorted out in order to correct any grammar errors that exist.

Response: Considering the Reviewer’s suggestion, we sort out the entire article, and a thorough spellchecking and proofreading by a native speaker is made.

  1. The similarity between FAIR1M and the data used in this article, such as UAVDT, needs to be explained in more detail.

Response: Considering the Reviewer’s suggestion, the similarities between FAIR1M and the data used in this article, such as UAVDT, are explained in section “4.1.1. Pre-training Datasets”.

  1. A more detailed description of the comparison of different algorithms is needed.

Response: Considering the Reviewer’s suggestion, more descriptions of the comparison of different algorithms has been improved in section “4.3. Experiment Results”.

  1. The conclusion section should be supported by results from comparisons, and the subsequent studies needs to be slightly clarified.

Response: Considering the Reviewer’s suggestion, the comparison results have been added and the subsequent studies are slightly clarified in the section “5. Conclusion”.

  1. Some pictures’ resolution are not high. Please make an improvement.

Response: Considering the Reviewer’s suggestion, the resolution of all the pictures in this paper are improved, especially ones with low resolution.

Reviewer 2 Report

The paper explores a proposed method for high-quality object detection in unmanned aerial vehicle (UAV) images. While the paper is well-detailed and provides an in-depth explanation, particularly in the literature review of state-of-the-art methods, I have significant concerns regarding the novelty of the work.

In my assessment, I do not see substantial novelty in the amalgamation of existing techniques, such as combining CNN with ViTs, employing existing optimizers, or pre-training a model with MIM. The only aspect that could be considered somehow "novel" is the authors' clever use of satellite images for pre-training, owing to certain similarities with UAV data.

Furthermore, despite the comprehensive nature of the manuscript, I found the text challenging to read. There were instances where the manuscript appeared verbose and indirect, attributable to the frequent use of the passive voice and lengthy sentences. I strongly recommend engaging a professional editor to enhance the overall quality of the manuscript.

In my opinion, while the paper presents a valuable application, it may not possess sufficient novelty to warrant publication in a journal of such high prestige.

Author Response

Response: Thanks very much for the reviewer's suggestions and comments. The main purpose of this paper is to achieve high performance of object detection in UAV images. Therefore, this paper draws on the concept of MetaFormer and designs a hybrid feature extraction backbone that combines CNN and ViT. By constructing and adding a relative position deviation module, the ability of ViT to obtain spatial information is improved, thereby improving the performance of the proposed backbone. In addition, a more efficient optimization method was utilized to further improve the performance of the model. On the basis of the above improvement and optimization, the proposed method can maintain a relatively low number of parameters while achieving better performance. Due to the difficulty of model training, this paper innovatively proposes using remote sensing image datasets that are much less than natural scene data to construct pre-trained model and has achieved good results.

Overall, this paper innovates and improves on the aspects of the backbone network, model optimization, and pre-trained process. It not only improves the object detection performance of UAV images, but also provides ideas for subsequent related research. Moreover, the proposed method has relatively few parameters and relies on consumer grade GPUs to achieve model training from scratch with less data and time, which can better meet the research needs of the academic community.

Considering the Reviewer’s suggestion, a thorough spellchecking and proofreading has been applied by a native speaker, and the overall quality of the manuscript has been improved.

Reviewer 3 Report

The authors present a hybrid backbone based on the combination of CNN and vision ViT, and applied MIM to aerospace remote sensing datasets to produce a pre-training model for the proposed method, which further improved UAV image object detection accuracy. However, there are still some details that need to be clarified, and the detailed suggestions are as follows:

1. In the experiments, the authors are recommended to add more comparisons with the latest methods.

2. The paper claims that SIMMIM learned better representations than MAE. However, there are many MIM methods, and please give more explanations about the reason for using SIMMIM.

3. In Figure 1, the method proposed in this study only uses features from S2 to S4 for transmission to the Encoder Layers, please give more explanations about the reason that only features from S2 to S4 are used for transmission to the Encoder Layers.

4. More descriptions and details should be given about the results of Table 5.

5. The entire article needs to be sorted out to ensure that there are no missing parts, such as formula 3.

6. The discussion section should establish the link between the major contributions and the results.

7. The bibliography should be updated by adding some more recent works, and it will be better to perform a thorough spellchecking and proofreading, preferably by a native speaker.

Author Response

  1. In the experiments, the authors are recommended to add more comparisons with the latest methods.

Response: Considering the Reviewer’s suggestion, we add several latest methods, SMT and BiFormer, which are proposed in this year, and the comparison results are shown in Table 4, Table 5, Figure 7, Figure 8, Figure 9, Figure 10, and Figure 11.

  1. The paper claims that SIMMIM learned better representations than MAE. However, there are many MIM methods, and please give more explanations about the reason for using SIMMIM.

Response: Considering the Reviewer’s suggestion, we give more explanations about the reason for using SIMMIM in the end of section “2.2. Masked Modeling Methods”.

  1. In Figure 1, the method proposed in this study only uses features from S2 to S4 for transmission to the Encoder Layers, please give more explanations about the reason that only features from S2 to S4 are used for transmission to the Encoder Layers.

Response: Considering the Reviewer’s suggestion, more explanations about the reason that only features from S2 to S4 are used for transmission to the Encoder Layers are given in the end of section “3.1. Overall Framework”.

  1. More descriptions and details should be given about the results of Table 5.

Response: Considering the Reviewer’s suggestion, more descriptions and details should be given about the results of Table 5.

  1. The entire article needs to be sorted out to ensure that there are no missing parts, such as formula 3.

Response: Thanks for the Reviewer’s suggestion, the entire article has been sorted out, and the missing parts have been supplemented.

  1. The discussion section should establish the link between the major contributions and the results.

Response: Considering the Reviewer’s suggestion, the discussion section have been improved.

  1. The bibliography should be updated by adding some more recent works, and it will be better to perform a thorough spellchecking and proofreading, preferably by a native speaker.

Response: Considering the Reviewer’s suggestion, some recent works, such as SMT and BiFormer, has been used in this paper, and the corresponding bibliographies have been added. In addition, a thorough spellchecking and proofreading has been applied by a native speaker.

Round 2

Reviewer 1 Report

The manuscript named “High-quality Object Detection Method for UAV Images Based on Improved DINO and Masked Image Modeling” has been reviewed. This work attempted to combine the SIMMIM and a hybrid backbone based on CNN and ViT to improved UAV image object detection accuracy. In general, the paper is well-written and well-organized, the results seem reliable.

The author has responded to all of the eight questions or small suggestions raised by the reviewers. The authors' responses are relatively accurate.

How, there are still several problems that need to be improved:

1. The current research status of the manuscript at home and abroad still needs to be strengthened, for example, the references are not comprehensive enough. It is recommended to cite a relatively relevant reference from this journal(https://doi.org/10.3390/rs15020440).

2. There are still a few grammar errors in the manuscript. For example, in the tense of abstracts, some should use the present tense and some should use the past tense, which can refer to some high-level writing literature published in this journal.

Suggest to give an acceptance after a minor revision.

Author Response

1. The current research status of the manuscript at home and abroad still needs to be strengthened, for example, the references are not comprehensive enough. It is recommended to cite a relatively relevant reference from this journal(https://doi.org/10.3390/rs15020440).
Response: Considering the Reviewer’s suggestion, we strengthen the manuscript, and cite some relatively relevant references.
2. There are still a few grammar errors in the manuscript. For example, in the tense of abstracts, some should use the present tense and some should use the past tense, which can refer to some high-level writing literature published in this journal.
Response: Considering the Reviewer’s suggestion, the existing grammar errors have been corrected.

Reviewer 2 Report

Despite the authors explanations, I still do not see enough novelty in the work to be published. I would like to leave the final decision to the editor. Other iterations from my side are not necessary.

Author Response

Thank you for the reviewers’ efforts and comments concerning our manuscript.

Reviewer 3 Report

Dear Authors,

I think the revised manuscript has solved my comments, and the current verison can be accepted.

Best wishes,

Author Response

(The authors gave the same response as above.)
